# TLR-induced STK25 activation promotes IRF5-mediated inflammation

Matthew R Rice[1,2], Bharati Matta[2], Loretta Wang[2], Qi Luo[3], Jeremy De Guzman[3], Dinesh Srinivasan[3] , Katelyn R Ludwig[4], Surya Indukuri[1,2], Leianna Brune[2], Seng-Lai Tan[3], Betsy J Barnes[1,2,5]

The transcription factor interferon regulatory factor 5 (IRF5) functions as an important mediator of the inflammatory response downstream of MyD88-dependent TLRs. Whereas dysregulation of IRF5 activity has been implicated in the development of numerous autoimmune diseases including systemic lupus erythematosus, the factors that modulate TLR-induced IRF5 post-translational modifications are poorly understood. The focus of this study was to identify novel kinases in TLR-MyD88-IRF5 signaling. We performed a kinome-wide siRNA screen in human THP-1 monocytic cells and identified serine/threonine protein kinase 25 (STK25) as a positive regulator of pro-inflammatory cytokine production via phosphorylation of IRF5 at Thr265, leading to IRF5 transcriptional activation. We further found that STK25 undergoes autophosphorylation in response to multiple TLR triggers. Findings were validated in *Stk25*-deficient primary immune cells revealing a significant attenuation in R848-induced IRF5 nuclear translocation and pro-inflammatory cytokine production. Finally, we detected increased levels of STK25 autophosphorylation in immune cells from systemic lupus erythematosus donors compared with healthy controls. These findings implicate STK25 as a new regulator of TLR7/8 signaling through the modulation of IRF5 activation.

## Introduction

The innate immune system uses pattern-recognition receptors to sense the presence of invading pathogens. TLRs are an important class of pattern-recognition receptors that recognize conserved microbial elements known as pathogen-associated molecular patterns and host-derived signals from injured cells called damage-associated molecular patterns (1). TLR engagement facilitates the activation of signaling pathways that culminate in the induction of pro-inflammatory cytokines and type I IFNs (1). The TLR-induced production of pro-inflammatory mediators promotes

microbial clearance and primes the secondary pathogen-specific adaptive immune response (1). The transcription factor interferon regulatory factor 5 (IRF5) functions as a critical modulator of the inflammatory response downstream of MyD88-dependent TLR activation (2, 3). The activation of IRF5 involves a series of post-translational modifications that include TRAF6-mediated ubiquitination and phosphorylation by IKKβ (4, 5, 6). In turn, a network of negative regulatory factors, including Lyn, IRF4, and IKKα, function to restrain the activation of IRF5 downstream of TLR ligation (7, 8, 9). Genetic polymorphisms within or near the *IRF5* locus have been associated with IRF5 hyperactivation and the dysregulation of IRF5 activity has been implicated in the pathogenesis of systemic lupus erythematosus (SLE) (10). Given that dysregulated functions of TLRs have also been implicated in SLE, and IRF5 is a critical downstream mediator of MyD88-dependent TLR signaling, the identification of kinases that regulate the TLR-mediated activation of IRF5 may lead to the development of therapeutic agents for the treatment of patients with autoimmune and inflammatory conditions.

Serine/threonine protein kinase 25 (STK25) is a member of the germinal center kinase (GCK)-III subfamily of the mammalian sterile 20-like (MST) kinase family. STK25 has been implicated in the regulation of metabolic homeostasis, neuronal polarization, cell migration, Golgi organization, apoptosis, and tumorigenesis (11, 12, 13, 14, 15). However, a role for STK25 in the modulation of TLR signaling has yet to be defined.

In this study, we screened for putative IRF5 kinases involved in the regulation of TLR7/8-mediated pro-inflammatory cytokine release. Of the candidate kinases identified, only STK25 directly phosphorylated IRF5 in vitro. STK25 promoted IRF5 transcriptional activation in HEK293T cells via phosphorylation of Thr265. In addition, signaling through TLR7/8 induced the expression and activation of STK25. Furthermore, STK25 positively regulated TLR7/9-induced IRF5 nuclear translocation and pro-inflammatory cytokine production in murine primary immune cells. Finally, PBMCs from patients with SLE demonstrated elevated expression of autophosphorylated STK25 as compared with healthy controls. Altogether, our findings support a

[1]Donald and Barbara Zucker School of Medicine at Hofstra/Northwell, Hempstead, NY, USA [2]Insitute of Molecular Medicine, The Feinstein Institutes for Medical Research, Manhasset, NY, USA [3]Hoffman La-Roche, Nutley, NJ, USA [4]Department of Integrative Physiology, University of Colorado Boulder, Boulder, CO, USA [5]Departments of Molecular Medicine and Pediatrics, Donald and Barbara Zucker School of Medicine at Hofstra/Northwell, Hempstead, NY, USA

Correspondence: bbarnes1@northwell.edu

role for STK25 in the regulation of TLR signaling and thereby implicate STK25 as a potential therapeutic target for the treatment of IRF5-mediated immunological disorders.

## Results

### Identification of positive regulators of TLR7/8-induced pro-inflammatory cytokine production

To identify candidate kinases involved in the regulation of IRF5 downstream of TLR7/8-mediated signaling, we conducted a kinome-wide siRNA screen in the THP-1 human monocytic cell line. Each protein was targeted by a single siRNA molecule at a time, with a total of four distinct siRNA constructs per target. The cells were stimulated with R848, a TLR7/8 agonist, 48 h after siRNA knockdown, and culture supernatants were harvested 24 h post-stimulation. Alterations in the R848-induced production of TNF-$\alpha$ following siRNA knockdown were determined by AlphaLISA immunoassay (Fig 1A). We used strictly standardized mean difference (SSMD) scores to compare the effect of each siRNA molecule on R848-induced pro-inflammatory cytokine production. An SSMD score of –1 for a given siRNA molecule was considered a positive hit and genes with two or more hits were selected for follow-up analysis. The identification of well-established TLR signaling components, interleukin-1 receptor-associated kinase 1 (IRAK1) and IRAK2 as positive regulators of the inflammatory response to R848, provided validation for the screen (Fig 1A). The 20 targets that exhibited the greatest reduction in R848-induced pro-inflammatory cytokine production following siRNA knockdown are ranked by SSMD score in Fig 1B. Given that siRNA knockdown of multiple kinases in THP-1 cells reduced the R848-induced production of pro-inflammatory cytokines, we sought to confirm the relevance of these findings in human primary myeloid cells. We isolated monocytes from the peripheral blood of multiple healthy donors and generated monocyte-derived dendritic cells (MDDCs) via incubation with GM-CSF and IL-4. The cells were subjected to targeted siRNA knockdown for 48 h and then stimulated with R848 for 24 h. The culture supernatants were harvested and TNF-$\alpha$ and IL-6 production was quantified via AlphaLISA immunoassay. SSMD scores for each siRNA construct were generated as before, factoring in the inhibition of TNF-$\alpha$ and IL-6 from multiple healthy donors to generate a robust, singular measure of siRNA efficiency. Interestingly, knockdown of MAP3K19, STK25, and SPHK2 in human primary monocytes led to a substantial reduction in the R848-mediated production of TNF-$\alpha$ and IL-6 (Fig 1C). In human primary MDDCs, each siRNA molecule used to target MAP3K19, STK25, SPHK2, and STK16 dramatically inhibited the R848-induced production of TNF-$\alpha$ and IL-6, with SSMD scores less than –1 (Fig 1D). Whereas many of the hits represent known modulators of TLR signaling, serine/threonine protein kinase 16 (STK16), STK25, mitogen-activated protein kinase kinase kinase 19 (MAP3K19), and sphingosine kinase 2 (SPHK2), were considered priority targets for mechanistic follow-up studies, as they had never been associated with TLR7/8-induced cytokine production.

### *STK25* is highly expressed in multiple immune cell types

Since STK16, STK25, MAP3K19, and SPHK2 have never been implicated in the regulation of TLR7/8-mediated cytokine production, we used a published RNA-seq dataset to analyze the gene expression levels of each kinase in sorted peripheral blood cell populations from healthy human donors (16). The expression levels of *STK16*, *MAP3K19*, and *SPHK2* were particularly low in most of the examined immune cell populations with a mean transcripts per million < 8 (Fig 1E–G). However, *STK25* was expressed in nearly all immune cell populations with a mean transcripts per million > 15, except for neutrophils (Fig 1H). We found that *STK25* was highly expressed in classical monocytes, a cell type that exhibits the highest expression of *IRF5* (Fig 1H and I). We next examined STK25 protein expression in a spectrum of human immortalized cell lines representative of B cells, T cells and monocytes, by immunoblot analysis and determined that STK25 was expressed in all cell types, with the highest expression in THP-1 monocytes (Fig 1J and K). In summary, STK25 was broadly expressed in human immune cells, suggesting a previously uncharacterized role for STK25 in the regulation of immune cell function. Furthermore, STK25 was highly expressed in human monocytes, a cell type that is critical to the innate immune response to pathogens.

### STK25 induces IRF5 transcriptional activation via phosphorylation of Thr265

Given the importance of IRF5 post-translational modifications in the modulation of TLR-induced transcriptional activation, we evaluated the ability of each target kinase to phosphorylate a biotinylated C-terminal construct (residues 222–467) of IRF5 via an in vitro scintillation proximity assay. The phosphorylation of IRF5 by IKK$\beta$ and IRAK1 provided validation for the assay (Fig S1A). Of the candidate kinases tested, only STK25 possessed the ability to phosphorylate the truncated form of IRF5 (Fig 2A). In addition, we found that IKK$\beta$ and STK25 could phosphorylate full-length IRF5 in a dose-dependent manner in vitro via an independent luminescent kinase assay system (Fig 2B–E). Finally, we evaluated the kinetics of STK25- and IKK$\beta$-mediated IRF5 phosphorylation in vitro by Phos-tag immunoblot analysis. Interestingly, whereas the kinetics of IRF5 phosphorylation by STK25 and IKK$\beta$ were similar, with detectable IRF5 phosphorylation within 30 min, and substantial phosphorylation by 60 min, the patterns of modification appeared different suggesting phosphorylation at distinct sites (Fig 2F). Since STK25 functions as a serine/threonine kinase, we sought to determine which IRF5 residues are specifically modified by STK25. Immunoblot analysis of 60 min kinase reactions with STK25 and IRF5 revealed that STK25 phosphorylated IRF5 at threonine (Thr) residues (Fig 2G). To identify the specific IRF5 residues that are phosphorylated by STK25, we used mass spectrometry. We found that STK25 phosphorylated IRF5 at multiple residues, including Thr183, Thr265, and Thr314 (Figs 2H and S1B). Since these IRF5 residues have not been previously implicated in the regulation of IRF5 activity, we aligned IRF5 protein sequences from characterized human isoforms and multiple species to evaluate conservation. Interestingly, Thr265 is completely conserved across all human isoforms and queried species, whereas Thr183 is conserved across most human isoforms

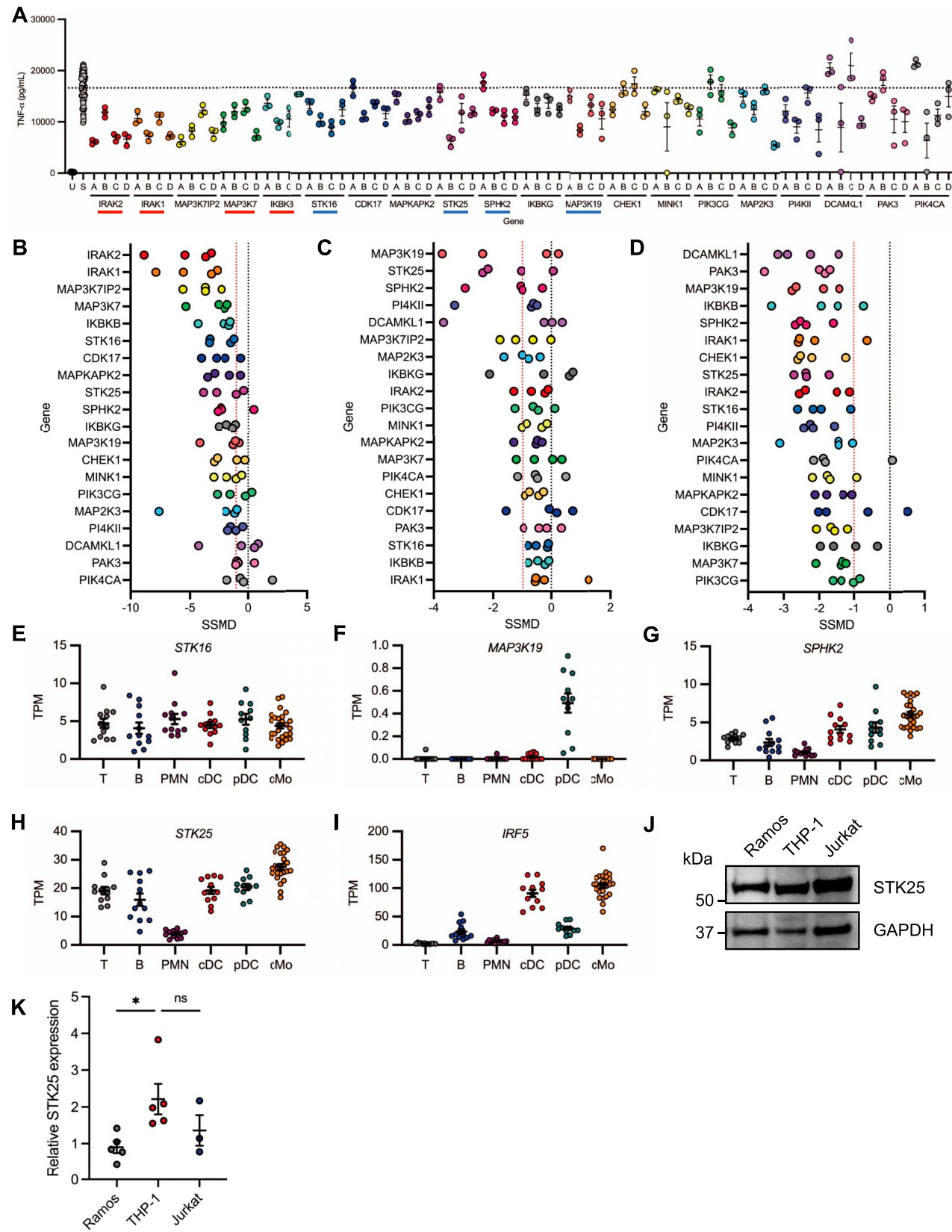

and substituted with an alanine residue in *M. musculus* (Figs 2I and J and S1C and D). The evolutionary conservation of Thr265 suggests that this residue may be important for IRF5 function. To validate our mass spectrometry data and directly confirm the ability of STK25 to phosphorylate IRF5 at Thr265, we first generated an alanine-substituted mutant form of IRF5 at residue 265 (IRF5-T265A) via site-directed mutagenesis. Then, we generated WT-IRF5 and IRF5-T265A recombinant proteins using an in vitro cell-free expression system. We confirmed the expression of WT-IRF5 and IRF5-T265A recombinant proteins by immunoblot analysis (Fig 2K). The WT-IRF5 and IRF5-T265A recombinant proteins were incubated with STK25 and subjected to an in vitro luminescent kinase assay. As expected, the total phosphorylation of IRF5-T265A by STK25 was significantly reduced compared with WT-IRF5 (Fig 2L). The residual STK25-mediated phosphorylation of IRF5-T265A is most likely due to the modification of additional residues, including Thr183 and Thr314 (Fig S1B). Moreover, immunoblot analysis of 60 min kinase reactions with STK25 and either WT-IRF5 or IRF5-T265A demonstrated a significant reduction in Thr-based phosphorylation of IRF5-T265A compared with WT-IRF5 (Fig 2M and N). Therefore, STK25 is a putative IRF5 kinase that phosphorylates IRF5 at a highly conserved residue, Thr265.

Whereas the phosphorylation of serine (Ser) residues in the C-terminal region of IRF5 has been demonstrated to regulate IRF5 activity (5, 6, 17), the functional significance of threonine phosphorylation is poorly understood. As such, we used an interferon-stimulated response element (ISRE) firefly luciferase (ISRE-Luc) reporter system to evaluate STK25-mediated IRF5 activation in HEK293T cells. We observed basal ISRE-Luc activity with the expression of WT-IRF5 alone (Fig 3A). However, the co-expression of WT-IRF5 and STK25 led to a significant increase in ISRE-Luc activity, indicating that STK25 promoted IRF5 transcriptional activity (Fig 3A). The expression of STK25 alone had a negligible effect on ISRE-Luc activity, so these findings are not simply due to the additive effects of WT-IRF5 alone and STK25 alone (Fig 3A). Interestingly, the IRF5-T265A mutant, which is resistant to STK25-mediated phosphorylation at Thr265, exhibited substantially reduced basal ISRE-Luc activity compared with WT-IRF5 (Fig 3A). Whereas the co-expression of IRF5-T265A and STK25 led to a slight increase in ISRE-Luc activity compared with IRF5-T265A alone, the level of activation was not significantly different from IRF5-T265A alone or WT-IRF5 alone (Fig 3A). Importantly, the observed differences in the activation of

WT-IRF5 and IRF5-T265A were not due to altered expression of the IRF5-T265A mutant (Fig 3B). To determine whether endogenous STK25 expression contributes to the basal ISRE-Luc activity observed with the expression of WT-IRF5 alone, we generated *STK25*-deficient (STK25-KO) HEK293T cells via CRISPR-Cas9-mediated gene editing. The efficiency of STK25 deletion was essentially 100%, as STK25 protein expression was not detected in the STK25-KO cells by immunoblot analysis (Fig 3C). Interestingly, loss of endogenous STK25 expression in HEK293T cells led to a significant reduction in WT-IRF5-induced activation of the ISRE-Luc reporter, further supporting a role for STK25 in the activation of IRF5 in cells (Fig 3D). This effect was not restricted to WT-IRF5, as STK25-KO cells exhibited a further reduction in IRF5-T265A-mediated activation of the ISRE-Luc reporter compared with WT cells (Fig 3D). Reminiscent of data in Fig 3A, co-expression of WT-IRF5 and STK25 in WT cells revealed a significant increase in ISRE-Luc activity compared with WT-IRF5 alone, and these levels were significantly reduced in STK25-KO cells, further highlighting the importance of endogenous STK25 in IRF5 activation (Fig 3D). Last, the level of reporter activity was not significantly different when IRF5-T265A and STK25 were co-expressed in WT or STK25-KO cells. Altogether, these findings support a role for Thr265 in STK25-mediated IRF5 transcriptional activation in cells.

### STK25 responds to TLR7/8 activation in THP-1 cells and regulates TLR-induced pro-inflammatory cytokine production in murine primary immune cells

All members of the GCK-III subfamily require the phosphorylation of a conserved threonine residue within the activation T-loop for full kinase activity (18). The phosphorylation of STK25 at Thr174 (T174) is hypothesized to be the result of a *trans*-autophosphorylation event that involves STK25 homodimerization (18). As such, we examined whether TLR7/8 activation induces STK25 autophosphorylation at Thr174 in THP-1 cells by immunoblot analysis. Despite the detection of basal levels of phospho-STK25 (T174) (p-STK25) in untreated cells, we observed an increase in p-STK25 expression at 30 min post-stimulation with R848 (Fig 4A and B). Importantly, R848-induced STK25 autophosphorylation occurred before IRF5 nuclear translocation, which has been previously detected at 2 h post-stimulation in THP-1 cells, amongst other cell types (19, 20, 21). Interestingly, STK25 activation was not restricted to TLR7/8 signaling in THP-1

**Figure 1. Identification of kinases involved in the regulation of R848-induced inflammation.**
**(A)** A comprehensive siRNA screen was conducted in THP-1 cells to identify kinases that modulate R848-mediated pro-inflammatory cytokine release. The 20 proteins that exhibited the most significant reduction in R848-induced TNF-α production following siRNA knockdown are displayed in ranked order from left to right. Previously characterized regulators of TLR signaling are underlined in red. Targets underlined in blue have never been associated with TLR7/8-induced pro-inflammatory cytokine production. The black dotted line represents the mean of the R848-stimulated control group (*N* = 63 biological replicates). Each letter represents an independent siRNA construct (*N* = 3 technical replicates). **(A, B, C, D)** The color scheme used in (A) is conserved in (B, C, D). **(B)** The strictly standardized mean difference (SSMD) scores for each siRNA construct used in the identification of key regulators of R848-induced TNF-α production in the THP-1 screen are shown in descending order of significance. An SSMD score of −1 for a particular siRNA molecule was considered a positive hit and targets with two or more hits were selected for validation in follow-up studies with human primary cells. The black dotted line represents an SSMD score of 0, and the red dotted line represents an SSMD score of −1. **(C, D)** Human primary monocytes (C) and monocyte-derived dendritic cells (D) were stimulated with R848 for 24 h following siRNA knockdown of targets identified in the primary screen. SSMD scores for each siRNA construct were determined using the degree of siRNA-mediated inhibition of R848-induced TNF-α and IL-6 production from cells isolated from two independent donors. **(E, F, G, H, I)** Relative mRNA expression of *STK16* (E), *MAP3K19* (F), *SPHK2* (G), *STK25* (H), and *IRF5* (I) in sorted peripheral blood leukocytes from healthy control donors (*N* = 11–26). Normalized transcripts per million values were obtained from GSE149050. T, T cells; B, B cells; PMN, neutrophils; cDC, conventional dendritic cells; pDC, plasmacytoid dendritic cells; cMo, classical monocytes. **(J)** Immunoblot analysis of STK25 protein expression in untreated human immortalized cell lines. Blots were probed with antibodies against STK25 and GAPDH. **(K)** Densitometric analysis of STK25 protein levels after normalization to the expression of GAPDH (*N* = 3–4 biological replicates). *P < 0.05, U, unstimulated, S, R848-stimulated, ns, not significant. Data represent mean ± SEM.

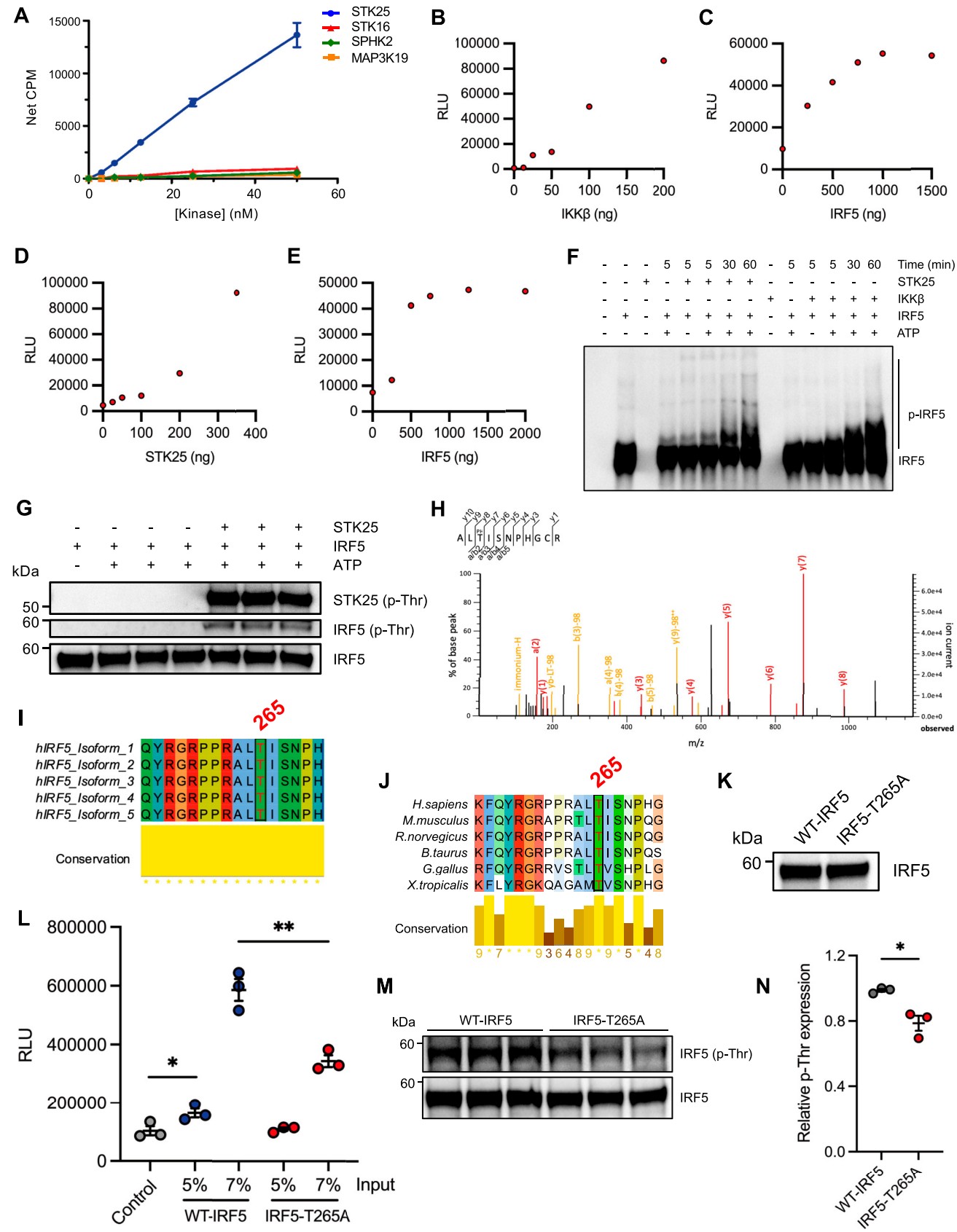

cells, as treatment with LPS, a TLR4 ligand, induced peak STK25 autophosphorylation at 6 h post-stimulation (Fig S2A and B). Moreover, the stimulation of Ramos B cells with CpG-B, a TLR9 ligand, induced robust autophosphorylation within 30 min, followed by a gradual decline over 24 h (Fig S2C and D). These data suggest that STK25 becomes activated downstream of multiple MyD88-dependent TLRs and may regulate additional intracellular signaling cascades in other immune cell types. We also found that STK25 protein expression was significantly up-regulated in THP-1 cells at 24 h post-stimulation with R848 (Fig 4C and D). Together, these findings indicate that STK25 is modulated at two levels (phosphorylation and protein expression) downstream of MyD88-dependent TLR signaling. To further characterize the role of STK25 in TLR-induced IRF5 activation in immune cells, we generated Stk25-deficient (Stk25$^{-/-}$) mice by Cre-mediated excision of exons 4 and 5 as previously described (12). Deletion of Stk25 was confirmed at the protein level by immunoblot analysis of Stk25$^{+/+}$ (WT) and Stk25$^{-/-}$ (KO) splenocytes (Fig 4E). Given the ability of STK25 to modulate R848-induced pro-inflammatory cytokine production in siRNA knockdown studies in human primary myeloid cells, we sought to further validate these findings via parallel experiments in immune cells from WT and KO mice. We isolated peripheral blood cells from WT and KO mice and evaluated IRF5 nuclear translocation at 2 h post-stimulation with R848 or CpG-B by multispectral imaging flow cytometry (19). We observed a significant reduction in R848-induced IRF5 nuclear translocation in KO peripheral blood CD11b$^+$ monocytes and B220$^+$ B cells compared with WT (Fig 4F and G). We also found a significant reduction in CpG-B-induced IRF5 nuclear translocation in KO peripheral blood B220$^+$ B cells relative to WT (Fig 4G). By intracellular flow cytometry, we observed a substantial reduction in the frequencies of IL-6$^+$ and TNF-$\alpha^+$ CD11b$^+$ splenocytes from KO mice compared with WT in response to an 18-h stimulation with R848 (Fig 4H and I). To evaluate the role of TLR specificity in STK25-mediated inflammatory responses, WT and KO splenocytes were stimulated with R848, CpG-B, or LPS for 24 h in vitro and the production of IL-6 in culture supernatants was quantified via ELISA. Notably, both R848- and CpG-B-induced production of IL-6 was significantly attenuated in KO splenocytes compared with WT (Fig 4J and K). However, LPS-induced production of IL-6, although reduced in KO splenocytes, was not found to be statistically different than WT (Fig 4L). Importantly, splenocytes

from WT and KO mice demonstrated similar viability on initial isolation and post-stimulation with various TLR ligands (Fig S2E–I). These findings supplement the results of our initial siRNA-based studies in human primary myeloid cells and provide additional insight into the conservation of STK25 function across multiple species and TLR signaling pathways.

### Basal autophosphorylation of STK25 at Thr174 is increased in PBMCs from patients with SLE

Whereas the activation of IRF5 plays a vital role in both the innate and adaptive arms of the immune system, IRF5 hyperactivation contributes to the development of autoimmunity (22, 23). In multiple murine models of SLE, inhibition of IRF5 attenuates disease activity, and thus IRF5 has emerged as an important therapeutic target (24, 25, 26). Given the ability of STK25 to induce the activation of IRF5, we sought to determine whether STK25 is autophosphorylated in immune cells from SLE patients at steady-state. To begin to address this hypothesis, we obtained PBMCs from a small cohort of healthy controls and patients with SLE and measured total STK25 and p-STK25 expression by immunoblot analysis. Interestingly, we observed a substantial increase in p-STK25 expression in SLE PBMCs compared with healthy controls, despite similar expression levels of total STK25, suggesting that STK25 is hyperactivated in SLE PBMCs (Fig 5A–C). Altogether, these findings implicate STK25 as a potential driver of IRF5 hyperactivation in SLE (Fig 5D).

## Discussion

In our efforts to further characterize regulators of IRF5 activation and function downstream of MyD88-dependent TLR signaling, we identified 4 new kinases – STK16, MAP3K19, SPHK2, and STK25 – that when knocked down, led to a reduction in the R848-induced production of TNF-$\alpha$ in THP-1 monocytes as well as IL-6 in human primary monocytes and MDDCs (Fig 1A–D). Interestingly, the degree of inhibition of R848-induced pro-inflammatory cytokine production was cell type-specific, with a more profound effect observed in MDDCs for all candidate kinases (Fig 1C and D). Whereas

**Figure 2. STK25 phosphorylates IRF5 at Thr265 in vitro.**
**(A)** Dose-dependent phosphorylation of a biotinylated C-terminal construct of IRF5 (residues 222–467, 3 $\mu$M) by candidate kinases in an in vitro scintillation proximity assay (N = 4 biological replicates). **(B, C, D, E)** Phosphorylation of full-length IRF5 by IKK$\beta$ (B, C) or STK25 (D, E) in an in vitro luminescent kinase assay. **(B)** Dose-dependent phosphorylation of IRF5 (500 ng) by IKK$\beta$. **(C)** Dose-dependent phosphorylation of IRF5 by a fixed amount of IKK$\beta$ (100 ng). **(D)** Dose-dependent phosphorylation of IRF5 (500 ng) by STK25. **(E)** Dose-dependent phosphorylation of IRF5 by a fixed amount of STK25 (350 ng). **(F)** Phos-tag immunoblot analysis of the kinetics of STK25- and IKK$\beta$-mediated phosphorylation of full-length IRF5 in vitro. Blots were probed with an antibody against total IRF5. p-IRF5, phosphorylated IRF5. **(G)** Immunoblot analysis of IRF5 phosphorylation at threonine residues following incubation with STK25 in an in vitro kinase assay for 1 h at RT. Blots were probed with antibodies against total p-Thr and IRF5. Representative of three independent experiments. p-Thr, phosphorylated threonine. **(H)** Mass spectrometry-based identification of Thr265 as an STK25-dependent IRF5 phosphorylation site. In vitro kinase reactions with IRF5 and STK25 were incubated for 1 h at RT, subjected to SDS–PAGE, and the gel was stained with Coomassie blue. Protein bands were excised, destained, and digested with trypsin. Representative of three independent experiments. **(I)** Conservation of Thr265 across multiple human isoforms of IRF5. **(J)** Conservation of Thr265 in IRF5 protein sequences from multiple species. **(K)** WT-IRF5 and IRF5-T265A were generated via an in vitro cell-free protein expression system and evaluated by immunoblot analysis. Blots were probed with an antibody against total IRF5. **(L)** Phosphorylation of WT-IRF5 and IRF5-T265A by STK25 in an in vitro luminescent kinase assay (N = 3 biological replicates). **(M)** Immunoblot analysis of WT-IRF5 and IRF5-T265A phosphorylation at threonine residues following incubation with STK25 in an in vitro kinase assay for 1 h at RT. Blots were probed with antibodies against total p-Thr and IRF5. **(N)** Densitometric analysis of WT-IRF5 and IRF5-T265A phosphorylation at threonine residues after normalization to the expression of total IRF5 (N = 3 biological replicates). *P < 0.05, **P < 0.01. Data represent mean ± SEM.

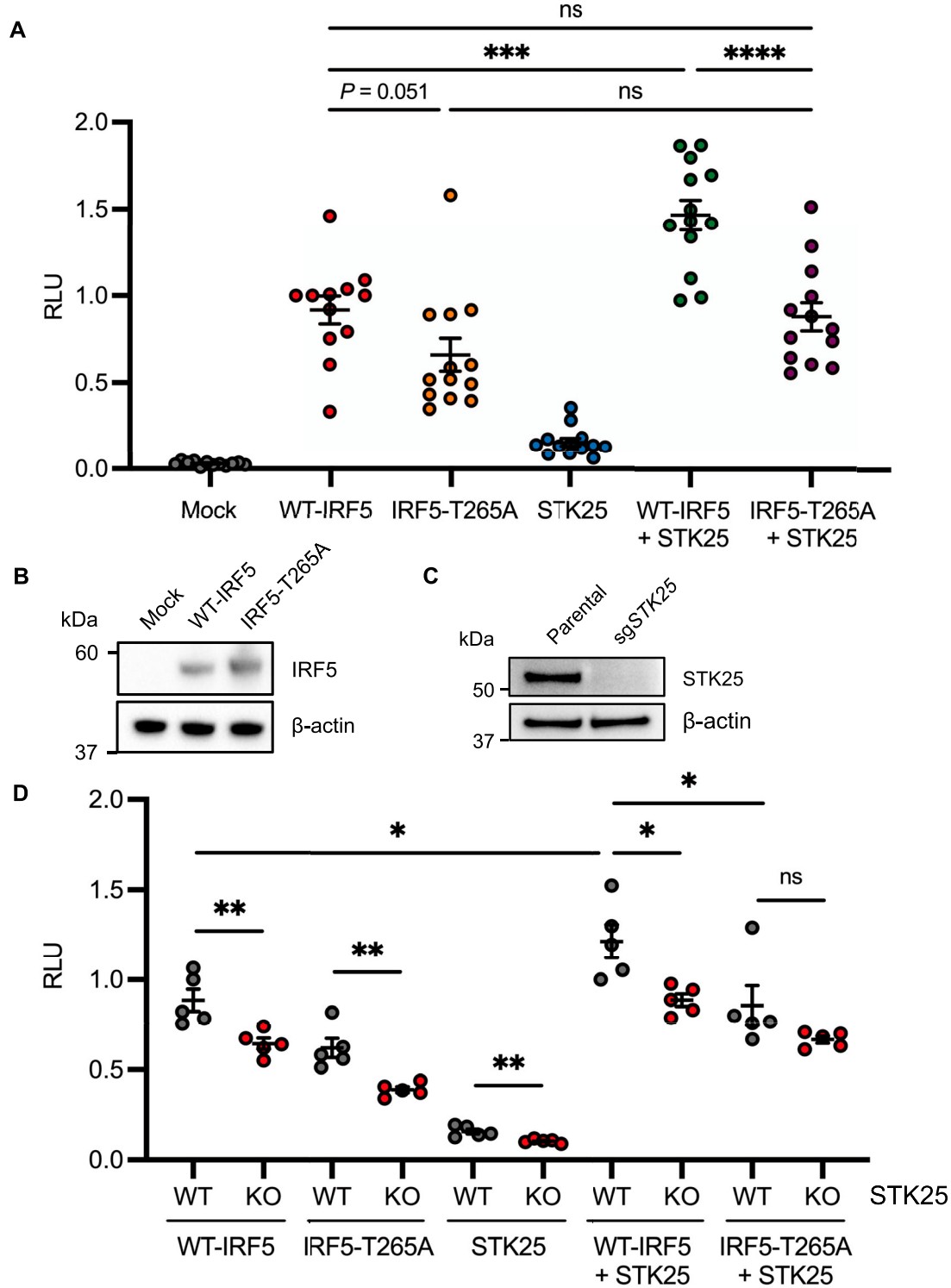

**Figure 3. STK25-mediated phosphorylation of IRF5 at Thr265 induces transcriptional activation in HEK293T cells.**
**(A)** HEK293T cells were co-transfected either alone or in combination with IRF5-FLAG, IRF5-T265A-FLAG, or STK25 in addition to an interferon-stimulated response element (ISRE) firefly luciferase (ISRE-Luc) reporter and a cytomegalovirus (CMV) *Renilla* luciferase (CMV-RL) internal control. The firefly luciferase signal was normalized to the *Renilla* luciferase signal to determine the relative light units (RLU). For each experiment, the relative ratio for each sample was normalized to a sample with IRF5-FLAG, ISRE-Luc, and CMV-RL (*N* = 12–13 transfections from three independent experiments). **(B)** Immunoblot analysis of HEK293T cells co-transfected with IRF5-FLAG or IRF5-T265A-FLAG. Blots were probed with antibodies against IRF5 and β-actin. **(C)** Immunoblot analysis of *STK25*-deficient HEK293T cells generated via CRISPR-Cas9-mediated gene editing. Blots were probed with antibodies against STK25 and β-actin. **(A, D)** WT and *STK25*-deficient (KO) HEK293T cells were co-transfected as in (A) to

the reason for the observed differences in the ability of these kinases to regulate pro-inflammatory cytokine responses in human primary monocytes and MDDCs is currently unknown, the cell type-specific expression pattern of each kinase may be involved. Interestingly, gene expression profiling of peripheral leukocytes from healthy donors indicated that *STK25* was highly expressed in multiple cell types, especially classical monocytes (Fig 1H). In humans, classical monocytes are a subset of monocytes that use TLRs to initiate the inflammatory response to pathogens (27). The expression of *IRF5* was also elevated in classical monocytes and the aberrant activation of IRF5 in monocytes has been implicated in the development of several autoimmune conditions, including SLE (10, 28). Furthermore, the examination of STK25 protein expression in human cell lines supported the gene expression profiles in human primary cells, as STK25 was highly expressed in THP-1 cells (Fig 1J and K). These findings agree with previous reports of STK25 expression in non-hematopoietic cell types, in that STK25 was ubiquitously expressed in most tissues (29). Altogether, we have demonstrated that STK25 is expressed in multiple immune cell subsets, with the highest levels of expression in TLR-responsive monocytes.

Since the activation of TLR7/8 by R848 results in the IKK$\beta$-dependent activation of both NF-$\kappa$B and IRF5, we examined whether the regulation of pro-inflammatory cytokine production by STK16, MAP3K19, SPHK2, and STK25 was a direct result of their ability to regulate IRF5 activation. We determined that only STK25 could phosphorylate IRF5 in a series of biochemical assays (Fig 2A–F). Thus, the ability of STK16, MAP3K19, and SPHK2 to mediate TLR-induced pro-inflammatory cytokine production may be due to their regulation of NF-$\kappa$B activity as opposed to IRF5. Alternatively, these kinases could potentially phosphorylate the N-terminal region of IRF5 between residues 1–221, as the screening assay was performed with a truncated form of IRF5 that only contained residues 222–467. Whereas the phosphorylation of IRF5 at serine residues within the serine-rich region (SRR) is important for the IKK$\beta$-mediated activation of IRF5, the phosphorylation of IRF5 at a tyrosine (Tyr) residue, Tyr172, by PYK2 has been shown to promote IRF5 transcriptional activity (30). Interestingly, Tyr172 is absent from isoform 1 of human IRF5, so the isoform-specific regulation of IRF5 by putative kinases should also be considered. Therefore, the ability of STK16, MAP3K19, and SPHK2 to phosphorylate full-length IRF5 should be further evaluated in future studies.

To extend our understanding of the regulation of IRF5 by STK25, we mapped STK25 target residues by mass spectrometry. We found that STK25 phosphorylated IRF5 at several threonine residues, including Thr183, Thr265, and Thr314 (Figs 2H and S1B). Interestingly, we also detected the STK25-induced phosphorylation of IRF5 at Tyr313 in two independent replicates; however, definitive localization could not be confirmed (Fig S1E). As a serine/threonine kinase, STK25 has been shown to catalyze the phosphorylation of serine and threonine residues, with a preference for serine residues (31). Despite the regulation of multiple cellular processes by STK25, only a few direct targets of

STK25 have been identified. In addition to autophosphorylation at Thr174, STK25 phosphorylates cerebral cavernous malformation 3 (CCM3) at Ser39 and Thr43 (32). Moreover, STK25 phosphorylates 14-3-3$\zeta$, Salvador homolog-1 (SAV1), large tumor suppressor 2 (LATS2), and cAMP-dependent protein kinase type I-alpha regulatory subunit (PRKAR1A) at Ser58, Thr26, Ser872, and Ser77/Ser83, respectively (13, 33, 34, 35). Given the importance of serine residues within the SRR of IRF5 that regulate its activity, it was intriguing that STK25 preferentially targeted threonine residues in IRF5 over serine residues (36, 37). Importantly, the STK25-dependent phosphorylation of IRF5 at Thr265 was robustly detected in all experimental replicates with conclusive localization (Figs 2H and S1B). To investigate the functional significance of phosphorylation at Thr265, we examined IRF5 protein sequences from several species. We found that Thr265 was conserved in all analyzed organisms and in all characterized human isoforms of IRF5 (Fig 2I and J). In contrast, Thr183 is replaced with an alanine residue in murine IRF5 and is not completely conserved across human isoforms of IRF5 (Fig S1C and D). Whereas the phosphorylation of IRF5 represents an important step in IRF5 activation, the TLR-mediated expression of pro-inflammatory cytokines requires the induction of transcriptional activity (2, 17). Notably, we found that in the absence of stimulation, STK25 promoted the transcriptional activity of IRF5 (Fig 3A) and mutation of Thr265 to alanine prevented the induction of IRF5 transcriptional activity by STK25 (Fig 3A and B). To our knowledge, this is the first report of kinase-mediated transcriptional activation of IRF5 via regulation of Thr265.

Although we have characterized the STK25-mediated activation of IRF5 downstream of TLR engagement, the upstream mechanisms that govern the activation of STK25 have never been examined. Autophosphorylation of STK25 at Thr174 represents an important step in kinase activation (18). Indeed, STK25 was originally identified as an oxidant stress response kinase due to its ability to become activated by reactive oxygen intermediates in Ramos B cells (38). Subsequent studies in rodent L6 myoblasts and human medulloblastoma D283MED-TrkA (MB-TrkA) cells revealed that STK25 undergoes autophosphorylation in response to TNF-$\alpha$ and nerve growth factor (NGF), respectively (31, 39). Whereas STK25 becomes activated in multiple cell types with a variety of stimuli, we have demonstrated that STK25 also undergoes autophosphorylation and activation in THP-1 cells and Ramos B cells in response to TLR4, TLR7/8, and TLR9 engagement (Figs 4A and B and S2A–D). Furthermore, we detected an increase in STK25 protein expression at 24 h post-stimulation with R848 in THP-1 cells (Fig 4C and D). The induction of STK25 protein expression by R848 supports the notion of a positive feedback loop that drives STK25 activity in the context of TLR activation. Together, these data suggest that STK25 becomes up-regulated and activated downstream of several MyD88-dependent TLRs and may regulate additional intracellular signaling pathways in other immune cell types. The mechanisms regulating STK25 expression and activation in the immune system are thus an active area of interest.

---

evaluate normalized ISRE-Luc reporter activity (*N* = 5 biological replicates per genotype). *$P < 0.05$, **$P < 0.01$, ***$P < 0.001$, ****$P < 0.0001$, ns, not significant. Data represent mean ± SEM.

**Figure 4. STK25 responds to TLR7/8 activation in THP-1 cells and regulates TLR-induced pro-inflammatory cytokine production in murine primary immune cells.**
**(A)** Immunoblot analysis of STK25 autophosphorylation at Thr174 in THP-1 cells stimulated with R848 for 0.5, 1, 2, 4, 6, or 24 h. Blots were probed with antibodies against p-STK25 (T174), STK25, and GAPDH. **(B)** Densitometric analysis of p-STK25 (T174) protein levels after normalization to total STK25 protein levels and the expression of GAPDH ($N = 2$ independent experiments). **(C)** Immunoblot analysis of STK25 expression in THP-1 cells stimulated with R848 for 24 h. **(D)** Densitometric analysis of STK25 protein levels after normalization to the expression of $\beta$-actin ($N = 4$ biological replicates). UT, untreated. **(E)** Immunoblot analysis of STK25 protein expression in $Stk25^{+/+}$ (WT) and $Stk25^{-/-}$ (KO) splenocytes. Blots were probed with antibodies against STK25 and GAPDH. **(F, G)** Imaging flow cytometry analysis of IRF5 nuclear translocation in peripheral blood CD11b+ monocytes (F) and B220+ B cells (G) from WT and KO mice following stimulation with R848 or CpG-B for 2 h ($N = 3$ biological replicates per genotype). **(H, I)** Flow cytometry analysis of the frequencies of IL-6+CD11b+ (H) and TNF-$\alpha$+CD11b+ (I) splenocytes from WT and KO mice following stimulation with R848 for 18 h ($N = 5–8$ biological replicates per genotype). **(J, K, L)** Quantification of IL-6 production in culture supernatants of WT and KO splenocytes stimulated with R848 (J), CpG-B (K), or LPS (L) for 24 h ($N = 4–5$ biological replicates per genotype). *$P < 0.05$, **$P < 0.01$, ***$P < 0.001$. ns, not significant. Data represent mean ± SEM.

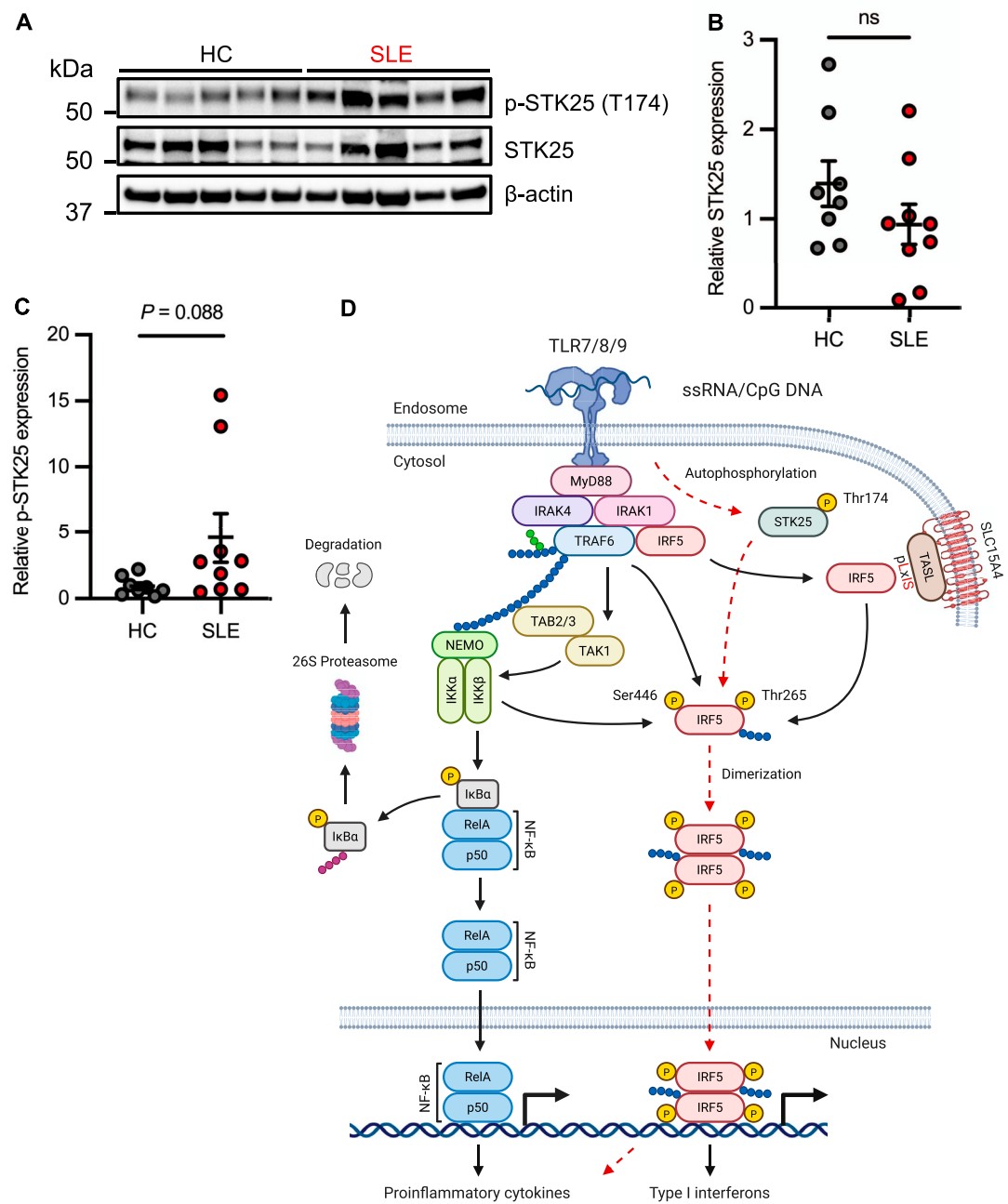

**Figure 5. Basal autophosphorylation of STK25 at Thr174 is increased in PBMCs from patients with systemic lupus erythematosus.**
**(A)** Immunoblot analysis of STK25 autophosphorylation at Thr174 in PBMCs from healthy donors and patients with systemic lupus erythematosus. Blots were probed with antibodies against p-STK25 (T174), STK25, and $\beta$-actin. **(B)** Densitometric analysis of total STK25 protein levels after normalization to the expression of $\beta$-actin ($N$ = 8–9 samples per condition). **(C)** Densitometric analysis of p-STK25 (T174) protein levels after normalization to total STK25 protein levels and the expression of $\beta$-actin ($N$ = 8–9 samples per condition). Data represent mean ± SEM. **(D)** Proposed model for STK25 as an IRF5 kinase downstream of TLR7/8/9 signaling (pathway highlighted by the red dashed arrows). The activation of TLR7/8 by ssRNA or TLR9 by CpG-B DNA induces the autophosphorylation of STK25 at Thr174 and stimulates the formation of the Myddosome, a signaling complex comprised of MyD88, IRAK1/4, TRAF6, and IRF5. The activation of IRAK1 by IRAK4 leads to the recruitment of TRAF6. The ubiquitination (colored circles) of TRAF6 provides a binding site for TAB2/3, thereby facilitating the activation of TAK1. NEMO, a component of the IKK complex, interacts with ubiquitinated TRAF6 to promote the activation of IKK$\beta$ by TAK1. IKK$\beta$-mediated phosphorylation of I$\kappa$B$\alpha$ induces its ubiquitination and degradation by the 26S proteasome. The inhibition of I$\kappa$B$\alpha$ permits the activation and nuclear translocation of the RelA and p50 subunits of NF-$\kappa$B. In the nucleus, NF-$\kappa$B regulates the expression of pro-inflammatory cytokines. In the alternate arm of the pathway, IRF5 undergoes TRAF6-catalyzed ubiquitination in addition to IKK$\beta$- and/or STK25-dependent phosphorylation at Ser446 and Thr265, respectively. The activation of IRF5 also involves an interaction with the adaptor protein, TASL, which binds to SLC15A4. Whereas IKK$\beta$ is required for TASL-mediated activation of IRF5, it is unknown if STK25 modulates the TASL-dependent pathway. Altogether, these modifications induce IRF5 dimerization and nuclear translocation. In the nucleus, transcriptionally active IRF5 modulates the expression of pro-inflammatory cytokines and type I interferons. Created with BioRender.com.

The canonical pathway of TLR-induced IRF5 activation involves phosphorylation, dimerization, and nuclear translocation (2, 17, 37) and we observed a significant reduction in both R848- and CpG-induced IRF5 nuclear translocation in KO peripheral blood monocytes and B cells, respectively, compared with WT (Fig 4F and G). In addition, we confirmed the importance of STK25 in TLR-mediated pro-inflammatory cytokine production through studies with KO splenocytes. Accordingly, the production of IL-6 downstream of R848 and CpG-B ligation was attenuated in KO splenocytes compared with WT (Fig 4J and K). These findings support our initial studies in human primary cells and highlight the ability of STK25 to regulate multiple steps involved in TLR-mediated IRF5 function, including upstream phosphorylation, nuclear translocation, transcriptional activation, and ultimately downstream pro-inflammatory cytokine production.

Throughout our studies, the mechanisms by which STK25 engages with IRF5 downstream of TLR7/8 activation have remained elusive. The association between a kinase and its substrate can be difficult to capture, due to the transient nature of the phosphorylation process. Indeed, IKK$\beta$ and IRF5 have yet to be shown to directly interact with each other (5, 6). However, the association between PYK2 and IRF5 prompted our investigation into whether STK25 binds to IRF5 (30). In blot overlay assays with human recombinant proteins, we detected a dose-dependent, albeit weak, interaction between IRF5 and STK25 (Fig S2J). In addition, we observed an intermediate dose-dependent interaction between IRF5 and IKK$\beta$, although binding between IRF5 and Lyn was more profound (Fig S2J). Yet, in immunoprecipitation studies in THP-1 cells and Ramos B cells, we were unable to observe an association between STK25 and IRF5. Given that the GCK-III subfamily members, MST3 and MST4, modulate signal transduction pathways through direct interactions with IKK$\beta$ and TRAF6, respectively, it is possible that STK25 is localized to IRF5 through an interaction with a TLR signaling component (40, 41). Alternately, the targeting of STK25 to IRF5 may depend on an unknown TLR adaptor protein. Accordingly, the adaptor protein, TASL, interacts with SLC15A4 to bind to IRF5, which facilitates the recruitment of IKK$\beta$ downstream of TLR engagement (42, 43, 44, 45). Whereas IKK$\beta$ modulates the TASL-mediated activation of IRF5, it is unknown if STK25 regulates this pathway in a similar manner (42, 43). Given the known functions of STK25 as a scaffold protein (14, 46), it is tempting to speculate that STK25 phosphorylates IRF5 through an interaction with the Myddosome and/or TASL/SLC15A4. Furthermore, studies will be required to address this mechanism.

Although activation of IRF5 in response to a microbial insult can aid in pathogen elimination, the dysregulation of IRF5 activity has been implicated in the pathogenesis of several autoimmune conditions. Specifically, *IRF5* genetic polymorphisms have been associated with aberrant leukocyte function in SLE through hyperactivation of IRF5 (20). Interestingly, the genetic ablation of *Irf5* has been found to confer protection in several murine models of SLE, including NZB/W F1, MRL/lpr, *Lyn*$^{-/-}$, and pristane-induced mice (24, 25, 26). As such, IRF5 has emerged as a therapeutic target for the treatment of SLE and other autoimmune conditions. Whereas the chemical inhibition of IRF5 remains an efficacious therapeutic strategy, the modulation of IRF5 kinases represents an important avenue for disrupting IRF5 function. The recent

development of small molecule inhibitors of STK25 provides an invaluable research tool and represents an important step toward the production of clinically relevant therapeutic agents (47). Even though STK25 has never been directly implicated in the pathogenesis of SLE, the ability of STK25 to regulate the TLR-induced activation of IRF5 indicates that STK25 may contribute to the hyperactivated IRF5 phenotype in this disease. Quite strikingly, we observed a substantial increase in p-STK25 expression in SLE PBMCs compared with healthy controls at steady-state, suggesting that STK25 is basally activated in SLE (Fig 5A–C). These data support the further examination of STK25 function in models of murine lupus. In conclusion, the identification of STK25 as a positive regulator of IRF5 in SLE could lead to the development of targeted therapeutic agents that could selectively inhibit IRF5 hyperactivation to reduce inflammation and disease burden.

# Materials and Methods

### Animals

*Stk25*-deficient (*Stk25*$^{-/-}$) mice were generated by Cre-mediated excision of exons 4 and 5 and genotyped as previously described (12). Deletion of *Stk25* was confirmed at the protein level by immunoblot analysis of *Stk25*$^{+/+}$ and *Stk25*$^{-/-}$ splenocytes (Fig 4E). As previously reported in the literature (12), *Stk25*$^{-/-}$ mice are viable, fertile, and healthy at homeostasis. All mice used in this study were between 8 and 12 wk of age. All animal care and experimental procedures were conducted in accordance with the *Guide for the Care and Use of Laboratory Animals* (National Academies Press, 2011) and approved by the IACUC of the Feinstein Institutes for Medical Research.

### Cell culture

THP-1 cells (TIB-202), Ramos cells (CRL-1596), Jurkat cells (TIB-152), and HEK293T cells (CRL-3216) were obtained from ATCC. Cells were certified mycoplasma-free at the time of receipt from ATCC. THP-1, Ramos, and Jurkat cells were maintained in RPMI 1640 medium supplemented with 10% FBS and 1X penicillin/streptomycin (P/S). HEK293T cells were maintained in DMEM (high glucose, GlutaMAX) supplemented with 10% FBS and 1X P/S. Cells were incubated at 37°C in a 5% CO$_2$ environment. For kinetic studies of p-STK25 (T174) expression, cells were seeded in 12-well plates at a density of $1 \times 10^6$ cells per well in 1 ml of complete RPMI 1640 and stimulated with R848 (1 $\mu$g/ml), CpG-B (2.5 $\mu$g/ml), or LPS (1 $\mu$g/ml) accordingly. For STK25 expression studies, cells were seeded in 12-well plates at a density of $1 \times 10^6$ cells per well in 1 ml of complete RPMI 1640 and stimulated with R848 (1 $\mu$g/ml) for 24 h.

### Generation of *STK25*-deficient HEK293T cells

The sgRNA for STK25 was obtained from IDT (Table 1). The target sequence was generated with the sgRNA designer tool from the Broad Institute. Recombinant Alt-R S.p. Cas9 nuclease (85 $\mu$g) was incubated with the target sgRNA (6 $\mu$l of 100 $\mu$M stock) for 20 min at

**Table 1. STK25 sgRNA target sequence used for CRISPR-Cas9-mediated gene editing.**

| sgRNA | Target sequence (5'-3') | PAM sequence |
|---|---|---|
| sgSTK25 | CATCGATAACCACACAAAGG | AGG |

RT. The reaction was supplemented with Alt-R Cas9 Electroporation Enhancer (4 $\mu$l) and combined with $4 \times 10^6$ cells resuspended in complete P3 nucleofection solution (71 $\mu$l) (V4XP-3024; Lonza). The samples were subjected to nucleofection using program CM-130 and maintained in DMEM (high glucose, GlutaMAX) supplemented with 10% FBS and 1X P/S. At 5 d post-nucleofection, $3 \times 10^6$ cells were collected to confirm knockout efficiency by immunoblot analysis.

### Kinome-wide siRNA screen

A kinome-wide siRNA library that contained 4 individually arrayed siRNA sequences in 384-well plates was purchased from QIAGEN. The library consisted of known kinases and associated proteins. To screen the library, 5 $\mu$l of each siRNA at 200 nM in Opti-MEM (31985062; Gibco) were added to 384-well plates followed by the addition of 5 $\mu$l of Lipofectamine 3000 RNAiMax Transfection Reagent (13778030; Invitrogen), at 12.5 $\mu$l/ml. After 30 min of incubation at RT, THP-1 cells were seeded at a density of $20 \times 10^4$ cells per well in 40 $\mu$l of RPMI 1640 supplemented with 1% FBS. After 48 h, the cells were stimulated with R848 (30 $\mu$M) for 24 h. Culture supernatants were harvested post-stimulation and the production of TNF-$\alpha$ was assessed using the AlphaLISA Immunoassay Kit (AL208C; Perkin-Elmer) according to the manufacturer's protocol. Cell viability was assessed using the CellTiter-Glo Luminescent Viability Assay (G7570; Promega). Efficiency of siRNA knockdown ranged from 40–80% between the four individual siRNAs per target.

### Human primary monocyte isolation

Human primary monocytes were isolated from healthy donors using the autoMACS Pan Monocyte Isolation Kit (130-096-537; Miltenyi Biotec). Briefly, one 40 ml Leukopak of blood was diluted with 100 ml of Dulbecco's phosphate-buffered saline (DPBS) + 2% FBS + 1 mM EDTA. A 35 ml aliquot of the diluted sample was transferred to a 50 ml ACCUSPIN tube (A2055; Sigma-Aldrich) containing 15 ml of Histopaque-1077 (10771; Sigma-Aldrich). After centrifugation, the interphase layer containing the PBMCs was transferred to a clean 50 ml tube and washed twice with DPBS + 2% FBS + 1 mM EDTA. Monocytes were isolated from the PBMC fraction through negative selection by depleting the cells labeled with CD3, CD7, CD16, CD19, CD56, CD123, and Glycophorin A using a MACS Separator (Miltenyi Biotec).

### Generation of human primary MDDCs

Human primary monocytes were seeded at a density of $5 \times 10^5$ cells per well in 1 ml of RPMI 1640 supplemented with 25 ng/ml of IL-4 and 50 ng/ml of GM-CSF. The media was replaced every 2 to 3 d. MDDCs were harvested after 7 d of differentiation.

### Human primary cell transfection

The siRNA constructs (QIAGEN) were mixed with 0.14 $\mu$l of Lipofectamine RNAiMAX Transfection Reagent (13778030; Invitrogen) and incubated at RT for 20 to 30 min. The siRNA (20 nM)/ lipofectamine (1.4 $\mu$l/ml) complex was then added to human primary monocytes or MDDCs that were seeded at a density of $5 \times 10^4$ cells per well or $2.5 \times 10^4$ cells per well in 100 $\mu$l of RPMI 1640, respectively. At 48 h post-transfection, the cells were stimulated with R848 (30 $\mu$M) for 24 h. Culture supernatants were harvested post-stimulation for analysis of pro-inflammatory cytokine production by AlphaLISA Immunoassay Kits.

### AlphaLISA immunoassay

The expression of IL-6 and TNF-$\alpha$ in culture supernatants was detected using the corresponding AlphaLISA Immunoassay Kit (AL223C and AL208C; PerkinElmer) according to the manufacturer's protocol.

### Immunoblot analysis

Briefly, whole cell lysates were prepared by lysing cells in NP-40 lysis buffer (50 mM Tris–HCl (pH 7.4), 150 mM NaCl, 1% NP-40 and 5 mM EDTA) (J60766-AP; Thermo Fisher Scientific) supplemented with Halt Protease Inhibitor Cocktail (87786; Thermo Fisher Scientific) and PhosStop Phosphatase Inhibitor Cocktail (4906845001; Roche). Sample protein concentrations were quantified using the DC protein assay (5000112; Bio-Rad). In general, 15-25 $\mu$g of protein per sample were separated by SDS–PAGE using the Bolt Bis-Tris system (Invitrogen). Proteins were transferred to 0.45 $\mu$m nitrocellulose membranes (SCNX8402XXXX101; MDI) using a wet tank transfer system. Transfer efficiency was assessed by incubating the membranes in 5 ml of Ponceau S (P7170; Sigma-Aldrich) for 5 min followed by destaining with Tris-buffered saline-0.05% Tween 20 (TBST) (J77500.K2; Thermo Fisher Scientific). The membranes were blocked for 1 h at RT with 5% BSA in TBST and incubated overnight at 4°C with the primary antibody diluted in the blocking buffer. The membranes were washed three times for 5 min each with TBST and incubated with the secondary antibody diluted in the blocking buffer for 1 h at RT. The membranes were washed three times for 5 min each with TBST and incubated with 1 ml of chemiluminescent detection reagent (RPN2232; Cytiva) for 3 min before image acquisition using a ChemiDoc MP Imaging System (Bio-Rad Laboratories). HRP-conjugated antibodies against $\beta$-actin (12620, 1:5,000; Cell Signaling) and GAPDH (3683, 1:5,000; Cell Signaling) were used as loading controls for protein normalization. Densitometric analysis was performed using the Image Lab software (Bio-Rad Laboratories).

### Immunoblot antibodies

Target (Vendor, Catalog Number, Primary Dilution, Secondary Dilution, Buffer).
 IRF5 (ab181553, 1:1,000, 1:10,000, 5% BSA/TBST; Abcam).
 p-Threonine (9386, 1:1,000, 1:10,000, 5% BSA/TBST; Cell Signaling).
 STK25 (ab157188, 1:1,000, 1:10,000, 5% BSA/TBST; Abcam).

STK25 + MST4 + MST3 (phospho T174 + T178 + T190) (ab76579, 1:1,000, 1:10,000, 5% BSA/TBST; Abcam).

$\beta$-actin HRP conjugate (12620, 1:5,000, n/a, 5% BSA/TBST; Cell Signaling).

GAPDH HRP conjugate (3683, 1:5,000, n/a, 5% BSA/TBST; Cell Signaling).

Secondary: IgG HRP conjugate (7074, n/a, 1:10,000, 5% BSA/TBST; Cell Signaling).

### Scintillation proximity assay (SPA)

Recombinant human IRF5 protein (residues 222–467; 3 $\mu$M) conjugated to biotin was incubated with recombinant human STK25, STK16, SPHK2, or MAP3K19 protein in the presence of isotopically-labeled ATP ($\gamma$-$^{33}$P) for 1 h. Samples were incubated with streptavidin-conjugated scintillation beads and $\gamma$-$^{33}$P incorporation was determined as described (48).

### In vitro kinase assay

Recombinant human IRF5 protein (ab173024; Abcam) (500 ng) was incubated with recombinant human STK25 protein (S43-10G-10; Signalchem) (350 ng) in kinase reaction buffer (40 mM Tris–HCl pH 7.4, 20 mM MgCl$_2$, 0.1 mg/ml BSA, 0.05 mM DTT, and 0.05 mM ATP) for 1 h at RT. The 10 $\mu$l reactions were quenched via the addition of 2X sample buffer, separated by SDS–PAGE, and transferred to nitrocellulose membranes. For the ADP-Glo kinase assays (TM313; Promega), 5 $\mu$l reactions were prepared with varying concentrations of recombinant human IRF5 protein (ab173024; Abcam) and recombinant human STK25 protein (S43-10G-10; Signalchem) or recombinant human IKK$\beta$ protein (I03-10BG-10; Signalchem) in kinase reaction buffer in 96-well plates. The reactions were incubated for 1 h at RT and quenched via the addition of 5 $\mu$l of ADP-Glo reagent to deplete the unconsumed ATP. The 10 $\mu$l reactions were incubated for 40 min at RT followed by the addition of 10 $\mu$l of kinase detection reagent to convert ADP to ATP. The 20 $\mu$l reactions were incubated for 30 min at RT and the luminescence of each reaction was measured with a plate-reading luminometer with an integration time of 1 s per well.

### Phos-tag immunoblot analysis

The Phos-tag immunoblotting method was described previously (49). Briefly, quenched kinase reactions (20 $\mu$l) were subjected to Phos-tag SDS–PAGE using a 7.5% SuperSep Phos-tag (50 $\mu$mol/liter) for 30 min at 15 mA and then 100 min at 30 mA. Gels were incubated three times in 25 ml of EDTA-containing transfer buffer (48.2 mM Tris, 38.9 mM glycine, 0.037% SDS, 20% methanol, 10 mM EDTA) for 20 min each with agitation. After washing the gels once for 20 min in EDTA-free transfer buffer (48.2 mM Tris, 38.9 mM glycine, 0.037% SDS, 20% methanol), proteins were transferred to 0.45 $\mu$m nitrocellulose membranes (SCNX8402XXXX101; MDI) using a wet tank transfer system. Transfer efficiency was assessed by incubating the membranes in 5 ml of Ponceau S (P7170; Sigma-Aldrich) for 5 min followed by destaining with TBST (J77500.K2; Thermo Fisher Scientific). The membranes were blocked for 1 h at RT with 5% BSA in TBST and incubated overnight at 4°C with the anti-IRF5 antibody diluted 1:

1,000 in the blocking buffer. The membranes were washed three times for 5 min each with TBST and incubated with IgG HRP conjugate secondary antibody diluted 1:10,000 in the blocking buffer for 1 h at RT. The membranes were washed three times for 5 min each with TBST and incubated with 1 ml of chemiluminescent detection reagent (RPN2232; Cytiva) for 3 min before image acquisition using a ChemiDoc MP Imaging System (Bio-Rad Laboratories).

### Mass spectrometry

Kinase reactions (10 $\mu$l) were briefly subjected to SDS–PAGE for 5 min using the Bolt Bis-Tris system (Invitrogen). The gel was stained with 25 ml of InstantBlue Coomassie Protein Stain (ab119211; Abcam) for 15 min at RT and protein bands of interest were excised. Gel slices were incubated with 1 ml of 30% ethanol for 20 min at 70°C. After three washes with 30% ethanol, the gel slices were stored in 1 ml of deionized H$_2$O at 4°C until downstream analysis. Gel slices were washed three times with 100 $\mu$l of 50 mM ABC/25% ACN (100 mM ammonium bicarbonate/25% acetonitrile) for 5 min each at 55°C with agitation. Gel slices were dehydrated with 100 $\mu$l of ACN. ACN was removed and the gel slices were dehydrated by vacuum centrifugation for 10 min. Gel slices were reconstituted in 50 $\mu$l of fresh 3 mM Tris (2-carboxyethyl)phosphine (TCEP)/50 mM ABC and incubated at 55°C for 20 min. Excess TCEP solution was removed and 50 $\mu$l of fresh 10 mM CEMTS/EtOH was added to the gel slices and subsequently incubated at 55°C for 20 min. Excess CEMTS solution was removed and the gel slices were washed three times with 100 $\mu$l of 50 mM ABC/25% ACN for 10 min at 55°C with agitation. Gel slices were dehydrated with 100 $\mu$l of ACN. ACN was removed and gel slices were dehydrated by vacuum centrifugation for 10 min. Gel slices were incubated with 5 $\mu$l of sequencing grade modified porcine trypsin (500 ng) in 50 mM ABC for 5 min on ice. Gel slices were incubated with 30 $\mu$l of 0.02% ProteaseMAX Surfactant in 50 mM ABC for 10 min on ice. Proteins were digested at 37°C overnight. Gel slices were incubated with 50 $\mu$l of 80% ACN/1% TFA for 15 min at 55°C with agitation. Samples were subjected to two elution steps with subsequent pooling and lyophilization by vacuum centrifugation. Peptide pellets were resuspended in 20 $\mu$l of Loading Buffer (5% DMSO, 0.1% TFA in water) and 3 $\mu$l of each sample was injected. Peptides were loaded on a 30 cm × 75 $\mu$m ID column packed with Reprosil 1.9 C18 silica particles (Dr. Maischt), and resolved on a 8–35% acetonitrile gradient in water (0.1% formate) at 250 nl/min. Eluting peptides were ionized by electrospray (2,200 V) and transferred into an Exploris 480 mass spectrometer (Thermo Fisher Scientific). Exploris 480 MS was fully recalibrated, with mass error <2 ppm. In nano-spray mode, >2,500 proteins and 15,000 peptides were identified (FDR <1%) from 200 ng of HeLa tryptic proteome. Nanoscale RP: <45 s peak width, >60 min effective elution during 90 min gradient. The water blank demonstrated no evident carry over. The MS was set to collect 120 K resolution precursor scans (m/z 380–2000 Th) and 60 K HCD fragmentation spectra at stepped 28, 33, 38% normalized energy, with first mass locked to 100 Th. Files were searched using the Mascot scoring function within ProteomeDiscoverer, with mass tolerance set at 5 ppm for MS1, and 0.01 D for MS2. Spectra were matched against the UniProt human consensus databases plus a database of common contaminants (cRAP). M-oxidation, N/Q-deamidation, and S/T/Y-phosphorylation

**Table 2. Primers used for site-directed mutagenesis of the pcDNA3.1+/C-(K)-DYK-IRF5 vector.**

| Primer | Sequence (5′-3′) |
| --- | --- |
| IRF5_T265A_F | CCGGGCCCTCGCCATCAGCAACC |
| IRF5_T265A_R | GGTGGCCGCCCCCGGTAC |

were set as variable modifications. Peptide-spectral matches were filtered to maintain FDR <1%.

## Reporter assay

HEK293T cells were seeded in white, tissue culture-treated 96-well plates at a density of $2.5 \times 10^4$ cells per well in 100 $\mu$l of DMEM and incubated at 37°C for 24 h. Cells were co-transfected with 100 ng of each target plasmid along with 100 ng of pGL3-ISRE-Luc and 20 ng of pRL-CMV using Lipofectamine 3000 Transfection Reagent (L3000015; Invitrogen). Cells were lysed 24 h post-transfection using the Dual-Glo Luciferase Assay System (E2920; Promega) and luminescence was analyzed using a plate reader. The firefly luciferase signal was normalized to the *Renilla* luciferase signal to determine the relative light units (RLU).

## Plasmids

The cDNA of human IRF5 isoform b (RefSeq accession no. NM_032643.4) was cloned into the pcDNA3.1+/C-(K)-DYK vector with a C-terminal FLAG tag (GenScript). The cDNA of human STK25 isoform 1 (RefSeq accession no. NM_001271977.2) was cloned into the pcDNA3.1+ vector (GenScript). The Q5 Site-Directed Mutagenesis Kit (NEB) was used to generate pcDNA3.1+/C-(K)-DYK-IRF5-T265A (Table 2). The pGL3-ISRE-Luc vector and the pRL-CMV vectors were obtained from Promega.

## In vitro cell-free protein expression

The TnT Quick Coupled Transcription/Translation System (L1171; Promega) was used to generate recombinant human proteins in vitro according to the manufacturer's protocol. Each reaction used 1 $\mu$g of plasmid DNA template, which consisted of pcDNA3.1+/C-(K)-DYK-IRF5 or pcDNA3.1+/C-(K)-DYK-IRF5-T265A, in a final volume of 50 $\mu$l. The reactions were incubated at 30°C for 90 min and the recombinant human protein products (5% or 7% of each reaction) were incubated with recombinant human STK25 protein (350 ng) for in vitro kinase reactions as previously described.

## Multiple sequence alignment

Protein sequences were extracted from the UniProt database and aligned with ClustalOWS and rendered in Jalview (50).

## Flow cytometry

Spleens from WT and $Stk25^{-/-}$ mice were harvested and homogenized with frosted slides in 4 ml of DPBS + 2% FBS. The samples were incubated with 5 ml of 1X RBC Lysis Buffer for 5 min on ice. Samples were washed with 5 ml of DPBS + 2% FBS and pelleted by centrifugation. Pellets were resuspended in 10 ml of DPBS + 2% FBS and filtered through a 70 $\mu$m cell strainer. The total number of cells in each sample was determined using a hemocytometer and adjusted to a final concentration of $4 \times 10^6$ cells/ml in complete RPMI 1640 medium. Cells were plated in an ultralow adherent 24-well plate and stimulated with R848 (1 $\mu$g/ml), CpG-B (2.5 $\mu$g/ml), or LPS (100 ng/ml) for 18 h. Cells were incubated with Brefeldin A for 2 h to facilitate the intracellular accumulation of IL-6. Cells were harvested, stained for surface markers in the dark, and subjected to fixation and permeabilization with 0.1% Triton X-100 in 2% PFA/PBS for 20 min. Cells were then stained for intracellular IL-6 and events were collected using the BD LSR Fortessa cell analyzer (BD Biosciences). Data were analyzed using the FlowJo Software (BD Biosciences).

## Antibodies

Target-Conjugate (Vendor, Catalog Number, Clone, Volume).
CD45-APC/Cy7 (103116, 30-F11, 0.25 $\mu$l; BioLegend).
CD11b-PE (101208, M1/70, 0.25 $\mu$l; BioLegend).
B220-BV510 (103248, RA3-6B2, 0.25 $\mu$l; BioLegend).
Ly6G-PerCP/Cy5.5 (127615, 1A8, 0.25 $\mu$l; BioLegend).
Ly6C-PE/Cy7 (128018, HK1.4, 0.25 $\mu$l; BioLegend).
CD4-BV421 (100438, GK1.5, 0.25 $\mu$l; BioLegend).
IL-6-APC (504508, MP5-20F3, 1 $\mu$l; BioLegend).
TNF-$\alpha$-FITC (506304, MP6-XT22, 1 $\mu$l; BioLegend).

## Imaging flow cytometry

Multispectral imaging flow cytometry was performed as previously described on the Amnis ImageStream (10, 19, 20, 28). Briefly, peripheral blood was isolated from WT and $Stk25^{-/-}$ mice via cardiac puncture and incubated with 5 ml of 1X RBC Lysis Buffer for 5 min on ice. Samples were washed with 5 ml of PBS, and the total number of cells in each sample was determined using a hemocytometer. Samples were adjusted to a final concentration of $0.5 \times 10^6$ cells/ml in complete RPMI 1640 medium. Cells were plated in a 24-well plate and stimulated with R848 (1 $\mu$g/ml) or CpG-B (2.5 $\mu$g/ml) for 2 h. Samples were washed with 5 ml of PBS, resuspended in 100 $\mu$l of PBS + 2% BSA, and stained for surface markers for 40 min at 4°C in the dark. Samples were washed with 5 ml of PBS and pellets were resuspended in 200 $\mu$l of 4% PFA/PBS. Samples were incubated for 1 h at RT and washed twice with 5 ml of PBS. For the permeabilization step, samples were resuspended in 300 $\mu$l of 0.5% Triton X-100 (200 $\mu$l) + 2 $\mu$l of Fc Shield (anti-mouse CD16/CD32, 70-0161-U500; Tonbo Biosciences) + 100 $\mu$l of 5% BSA/PBS and incubated overnight at 4°C in the dark. The next morning, samples were washed with 5 ml of 0.1% Triton X-100/PBS and incubated with 1 $\mu$l of IRF5 antibody (181553; Abcam) in 100 $\mu$l of 0.1% Triton X-100/PBS for 2 h at RT. Samples were washed three times with 5 ml of 0.1% Triton X-100/PBS and incubated with 2 $\mu$l of anti-rabbit IgG secondary-APC (A-10931; Invitrogen) in 100 $\mu$l of 0.1% Triton X-100/PBS for 40 min at RT. Samples were washed twice with 5 ml of 0.1% Triton X-100/PBS and nuclei were stained with DAPI (D1306; Invitrogen) for 5–8 min at RT. Samples were

washed once with 10 ml of PBS, resuspended in 50 $\mu$l of 4% PFA/PBS, and transferred to 1.5 ml microcentrifuge tubes. Events were collected using the Cytek Amnis ImageStream Mk II imaging flow cytometer (Cytek Biosciences). Data were analyzed using the IDEAS Image Analysis Software (Cytek Biosciences).

## Antibodies

Target-Conjugate (Vendor, Catalog Number, Clone, Volume).
   B220-FITC (553088, RA3-6B2, 2 $\mu$l; BD Pharmingen).
   CD4-BV510 (100559, RM4-5, 0.7 $\mu$l; BioLegend).
   CD8a-APC/Cy7 (100714, 53-6.7, 1 $\mu$l; BioLegend).
   CD11b-PE (101208, M1/70, 1 $\mu$l; BioLegend).
   CD11c-PE/Cy7 (117318, N418, 1 $\mu$l; BioLegend).
   Anti-rabbit IgG-APC (A-10931, Polyclonal, 2 $\mu$l; Invitrogen).

## ELISA

Splenocytes from WT and $Stk25^{-/-}$ mice were plated at a density of $4 \times 10^6$ cells/ml in 24-well plates and stimulated with R848 (1 $\mu$g/ml), CpG-B (2.5 $\mu$g/ml), or LPS (100 ng/ml) for 24 h. The expression of IL-6 in culture supernatants was detected using the Murine IL-6 Mini ABTS ELISA Development Kit (900-M50; PeproTech) according to the manufacturer's protocol.

## Blot overlay assay

The blot overlay assay protocol was described previously and modified accordingly (51). Various quantities of the bait protein (50-100 ng) were spotted on 1″ x 1″ sections of 0.2 $\mu$m nitrocellulose membranes (SCNX8401XXXX101; MDI) and incubated for 5–10 min at RT until the membranes were dry. The membranes were incubated in blocking buffer (2% non-fat milk in PBST) for 30 min at RT. The membranes were incubated with 100 ng of the overlay protein in the blocking buffer for 1 h at RT. The protein solution was discarded and the membranes were washed three times for 5 min each with blocking buffer. The membranes were incubated with anti-overlay protein antibody at 1:1,000 dilution in blocking buffer for 1 h at RT. The antibody solution was removed and the membranes were washed three times for 5 min each with blocking buffer. The membranes were incubated with IgG HRP conjugate secondary antibody at 1:2,000 dilution in blocking buffer for 1 h at RT. The antibody solution was removed and the membranes were washed three times for 5 min each with blocking buffer. The membranes were washed once for 5 min with PBS (pH 7.4) and processed according to the "Immunoblot analysis" protocol.

## Blot overlay proteins

Protein (Vendor, Catalog Number).
   Recombinant human STK25 (S43-10G-10; Signalchem).
   Recombinant human IRF5 (ab173024; Abcam).
   Recombinant human IKK$\beta$ (I03-10BG-10; Signalchem).
   Recombinant human Lyn (L13-10G; Signalchem).

## Blot overlay antibodies

Target (Vendor, Catalog Number, Primary Dilution, Secondary Dilution, Buffer).
   IRF5 (A303-386A, 1:1,000, 1:2,000, 2% non-fat milk/PBST; Bethyl Laboratories).
   Secondary: IgG HRP conjugate (7074, n/a, 1:2000, 2% non-fat milk/PBST; Cell Signaling).

## Statistical analysis

The statistical measure of SSMD was used to evaluate positive hits from the siRNA screen for confirmatory studies. For the primary screen in THP-1 cells, the robust version of SSMD was used on a plate-by-plate basis using the siRNA sequences on that plate as the negative reference. For the confirmatory screen in human primary cells, additional negative control siRNAs were included as the negative reference. A two-tailed $t$ test was used for comparisons between two samples with normal distribution. GraphPad Prism 9 (GraphPad Software) was used for statistical analysis and figure generation. A $P$-value of < 0.05 was considered statistically significant.

# Data Availability

Data supporting the conclusions are available within the article and supplementary material. In the case of the initial high-throughput kinome screen in THP-1, this is available within Table S1 and validation within primary myeloid cells is related to further ongoing and future studies and therefore will be available upon reasonable request. Phosphoproteomics sequencing data are contained within Table S2. Data generated or analyzed from imaging flow cytometry are available upon request.

# Supplementary Information

# Acknowledgements

The authors thank Dr. Gang Chen (Roche and EMD Serono) for helpful discussions. The authors would also like to acknowledge early contributions from Dr. Jaspreet Banga and Dr. Cherrie Thompson on the characterization of STK25 expression and function. We would also like to thank Ms. Margaret LaPan for the generation and breeding of $Stk25^{-/-}$ mice and Zarina Brune for her efforts to characterize STK25-deficient THP-1 cells. Last, we thank Dr. Paolo Cifani and Dr. Darryl Pappin of the Mass Spectrometry Shared Resource at Cold Spring Harbor Laboratory for their efforts on the phosphopeptide mapping studies. This work was supported by grants from the National Institutes of Health NIAMS 1 R01 AR 076242-03, Department of Defense CDMRP LRP W81XWH-18-1-0674, and Lupus Research Alliance to BJ Barnes. MR Rice was supported by Training Grant T32AI155392 in Translational Immunology.

## Author Contributions

MR Rice: conceptualization, data curation, formal analysis, investigation, methodology, and writing—original draft, review, and editing.
B Matta: formal analysis, supervision, investigation, and methodology.
L Wang: formal analysis, investigation, visualization, and methodology.
Q Luo: data curation, formal analysis, investigation, and methodology.
J De Guzman: data curation, formal analysis, investigation, and methodology.
D Srinivasan: data curation, formal analysis, investigation, and methodology.
KR Ludwig: formal analysis, investigation, and methodology.
S Indukuri: investigation.
L Brune: formal analysis and investigation.
S-L Tan: conceptualization and writing—review and editing.
BJ Barnes: conceptualization, resources, formal analysis, supervision, funding acquisition, visualization, project administration, and writing—original draft, review, and editing.

## Conflict of Interest Statement

The authors declare that they have no conflict of interest.

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
