## [Reviewer comments · Life Science Alliance]

Life Science Alliance

TLR-induced STK25 activation promotes IRF5-mediated inflammation

Matthew Rice, Bharati Matta, Loretta Wang, Qi Luo, Jeremy De Guzman, Dinesh Srinivasan, Katelyn Ludwig, Surya Indukuri, Leianna Brune, Seng-Lai Tan, and Betsy Barnes

DOI: <https://doi.org/10.26508/lsa.202503343>

Corresponding author(s): *Betsy Barnes, Feinstein Institute for Medical Research*

Review Timeline:

Submission Date:	2025-04-04
Editorial Decision:	2025-06-05
Revision Received:	2025-06-20
Accepted:	2025-06-27

Scientific Editor: *Sarita Hebbar*

Transaction Report:

June 5, 2025

RE: Life Science Alliance Manuscript #LSA-2025-03343-T

Betsy J Barnes
Feinstein Institute for Medical Research
Center for Autoimmune and Musculoskeletal Diseases
350 Community Dr.
350 Community Dr.
Manhasset 11030

Dear Dr. Barnes,

Thank you for submitting your manuscript entitled "TLR-induced STK25 activation promotes IRF5-mediated inflammation".

This work was evaluated by two expert reviewers whose comments are appended below. Both reviewers have commented that the study is well-designed, and adds a significant advance to the field. We would be happy to publish your paper in Life Science Alliance pending resolution of the reviewers' requests and final changes necessary to meet our formatting guidelines.

- Please be sure that the authorship listing and order is correct
- Please upload all figure files as individual ones, including the supplementary figure files; all figure legends should only appear in the main manuscript file
- Please add a Summary Blurb/Alternate Abstract in our system
- Please add a Category and Keywords for your manuscript in our system
- Please add the X and Bluesky handles of your host institute/organization, as well as your own and/or one of the authors, in our system
- Please add Author Contributions to our system as well
- Please add your main, supplementary figure, and table legends to the main manuscript text after the references section;
- Kindly include a 'Data Availability' statement

A. FINAL FILES:

B. MANUSCRIPT ORGANIZATION AND FORMATTING:

Sincerely,

Sarita Hebbar, PhD
Scientific Editor
Life Science Alliance
<http://www.lsajournal.org>

Reviewer #1 (Comments to the Authors (Required)):

This study reveals a role for serine/threonine protein kinase 25 (STK25) as a kinase responsible for (novel) threonine phosphorylation and activation of the IRF5 transcription factor downstream of signaling from TLR7/8 to drive proinflammatory cytokine responses in immune cells. STK25 emerged as a candidate from a kinome siRNA screen and it was subsequently verified as a kinase for IRF5. The conserved IRF5 threonine residues phosphorylated by STK25 were identified using multiple rigorous approaches including mass spec and mutagenesis and use of reporter cells lines demonstrated STK25-enhanced IRF5 transcriptional activity. Functional studies revealed induced expression and phosphorylation of STK25downstream of multiple TLRs and cell lines as well as cells from an STK25 KO mouse helped demonstrate reduced proinflammatory cytokine production after SLK25 depletion although in vivo only those downstream of intracellular TLRs were affected. Finally, analysis of PBMCs from SLE patients was suggestive of hyperphosphorylation of STK25 associated with SLE, in keeping with IRF5's known role in autoimmunity.

Overall this presents as a meticulously executed study which derives impact from establishing STK25 as a novel immune kinase with a role in IRF5 transcription downstream of TLRs, for the regulation of inflammatory responses. Given the potential for IRF5 as a therapeutic target in autoimmunity, STK25 now assumes similar potential. The study has employed an impressive range of biochemical and genetic approaches throughout including generation of an STK25 KO mouse although this model was used sparingly in the current study, one assumes it will be employed for disease studies in the future. Other cell biological and biomedical questions arise from these findings, perhaps for future exploration, but in its current form the study is a substantial and well defended advance in the field.

Minor comments:

1. What level of gene or protein knockdown was achieved for the siRNA treatments in the screen in Figure 1? While it is not necessary to show exhaustive figures for the initial screen, a general idea of the siRNA efficacy would be helpful.
2. No information is provided about characterization of the STK25 KO mouse line generated as part of this study. Blotting confirms loss of the STK25 protein but some comment on the presence or lack of any other phenotypic features is warranted given that STK25 kinase activity has been implicated in multiple other cellular contexts.

3. The model in Fig 5 is comprehensive, drawing from the data in this study as well as projections from the literature. It might be helpful to highlight the specific parts of this pathway that reflect the direct results of this study.

Comments:

The results show that TLR-induced phospho-STK25 precedes IRF5 nuclear translocation but multiple approaches failed to definitively pinpoint a direct interaction of STK25 and IRF5. The possible recruitment of TLR/MyD88-induced phospho-STK25 and IRF5 to endosomal membranes, possibly as part of an extended Myddosome, could be explored in the future by immunostaining or with tagged recombinant proteins to provide another line of evidence for STK25-IRF5 interaction and demonstrate the relevant location and molecular complex.

A broader examination of additional cytokines would be warranted to determine whether only proinflammatory cytokines are regulated (in 'normal' inflammation and disease) and whether some of the more tissue-specific cytokines are regulated.

Reviewer #3 (Comments to the Authors (Required)):

The manuscript by Rice et al identifies STK25 as a kinase that phosphorylates IRF5 and promotes pro-inflammatory cytokine induction downstream of TLR7/8. STK25 was identified in a kinome-wide siRNA screen assessing IL-6 as a readout. Validation of the top 20 targets in human monocyte derived dendritic cells confirmed that knockdown of STK25 and 3 other kinases dramatically reduced IL6 and TNF production downstream of R848. STK25 was highly expressed in different immune cells, including monocytes. Each target kinase was then assessed for its ability to phosphorylate IRF5 CTD by a number of methods, identifying STK25 as a kinase that could directly phosphorylate T265. STK25 could enhance IRF5 driven ISRE activity in HEKs where IRF5 T265A + STK25 did not and knockdown of STK25 resulted in reduction of IRF5-driven ISRE. Tolts indicate a role for STK25 in regulating IRF5 activity. STK25^{-/-} mice showed reduced frequencies of IL6 and TNF expressing CD11b⁺ cells and a reduction in IL6 in response to TLR7/8, TLR9 stimulation. Finally they showed that p-STK25 and IRF5 levels are higher in SLE PBMCs compared to controls, indicating a potential role for STK25 in driving hyper activation of IRF5.

This is a robust set of experiments that support the main points of the paper. The application of a number of complimentary ways to address the kinases and their targets was elegant. There were no major changes in the data necessary and although it would have been nice to see the effect of loss of STK25 in the resiquimod model of SLE, this data is not required to support publication of LFA given its scope. Minor comments include:

Justification for looking at the ability of STK25 to directly phosphorylate IRF5 is unclear. Did the authors conduct phosphosite prediction tool on potential targets such as NetPhos or other site? Or was it based on the known biology downstream of the receptors. This should be clarified.

The discussion is largely repetitive in that it restates each figure finding prior to discussing it. This should be modified so that the authors discuss some of the more salient aspects of STK25 biology - ie its ability to phosphorylate LATs and what that means in the context of TLR7/8 signaling.

Otherwise this is an excellent manuscript that should be accepted.

June 10, 2025

Dear Dr. Hebbbar,

We thank you and the Reviewers for their positive and constructive comments. Point-by-point responses are provided below. We appreciate that addressing these suggestions enable us to further strengthen our manuscript. All updates to the manuscript are highlighted in red.

Reviewer 1:

Summary - Overall this presents as a meticulously executed study which derives impact from establishing STK25 as a novel immune kinase with a role in IRF5 transcription downstream of TLRs, for the regulation of inflammatory responses. Given the potential for IRF5 as a therapeutic target in autoimmunity, STK25 now assumes similar potential. The study has employed an impressive range of biochemical and genetic approaches throughout including generation of an STK25 KO mouse although this model was used sparingly in the current study, one assumes it will be employed for disease studies in the future. Other cell biological and biomedical questions arise from these findings, perhaps for future exploration, but in its current form the study is a substantial and well defended advance in the field.

We thank the Reviewer for their interest in the topic and recognition of advancement in the field. Indeed, we are fully immunophenotyping the *Stk25*^{-/-} mice and are studying them in the context of two distinct models of murine lupus.

Minor comments:

1. What level of gene or protein knockdown was achieved for the siRNA treatments in the screen in Figure 1? While it is not necessary to show exhaustive figures for the initial screen, a general idea of the siRNA efficacy would be helpful.

We thank the Reviewer for their question. This information is now included in the Methods section under Kinome-wide siRNA screen. In general, we found a 40-80% level of knockdown between the individual siRNAs per target.

2. No information is provided about characterization of the STK25 KO mouse line generated as part of this study. Blotting confirms loss of the STK25 protein but some comment on the presence or lack of any other phenotypic features is warranted given that STK25 kinase activity has been implicated in multiple other cellular contexts.

We thank the Reviewer for the question. We now include a statement in the Methods section under Animals that *Stk25*^{-/-} mice are viable, fertile, and healthy at homeostasis, which has previously been reported by others. However, it is in the context of different disease models that have been induced on

the *Stk25*^{-/-} background where phenotypes are observed. Given that STK25 has never been studied in the immune system, this is our current area of interest, which provides a compelling direction for future work.

3. The model in Fig 5 is comprehensive, drawing from the data in this study as well as projections from the literature. It might be helpful to highlight the specific parts of this pathway that reflect the direct results of this study.

We thank the Reviewer for their suggestion and now highlight the parts of the pathway with dashed red arrows that are specifically related to the current study findings.

Comments:

The results show that TLR-induced phospho-STK25 precedes IRF5 nuclear translocation but multiple approaches failed to definitively pinpoint a direct interaction of STK25 and IRF5. The possible recruitment of TLR/MyD88-induced phospho-STK5 and IRF5 to endosomal membranes, possibly as part of an extended Myddosome, could be explored in the future by immunostaining or with tagged recombinant proteins to provide another line of evidence for STK25-IRF5 interaction and demonstrate the relevant location and molecular complex.

We absolutely agree with the Reviewer. This is certainly an area of interest that we are pursuing as STK25 has been also reported to act as a scaffold as well as a kinase.

A broader examination of additional cytokines would be warranted to determine whether only proinflammatory cytokines are regulated (in 'normal' inflammation and disease) and whether some of the more tissue-specific cytokines are regulated.

We agree with the Reviewer and are pursuing this broader characterization of STK25-mediated cytokine expression in our mouse models of disease. For the current study, we were focused on cytokines that are known to be downstream targets of IRF5.

Reviewer 3:

Summary - This is a robust set of experiments that support the main points of the paper. The application of a number of complimentary ways to address the kinases and their targets was elegant. There were no major changes in the data necessary and although it would have been nice to see the effect of loss of STK25 in the resiquimod model of SLE, this data is not required to support publication of LFA given its scope.

We thank the Reviewer for their interest in the study and thoughtful comments. We are indeed moving forward to characterize *Stk25*^{-/-} mice in two distinct models of murine lupus, one of which is TLR7-driven. Results will be reported in an independent manuscript.

Minor comments:

Justification for looking at the ability of STK25 to directly phosphorylate IRF5 is unclear. Did the authors conduct phosphosite prediction tool on potential targets such as NetPhos or other site? Or was it based on the known biology downstream of the receptors. This should be clarified.

We appreciate the Reviewers' comments. As stated in the first paragraph of the Introduction and beginning of the Results section (as well as second paragraph of the Discussion), we performed the kinome screen to identify kinases that regulate TLR-mediated activation of IRF5. This was the primary focus of the study. While the initial downstream readout of the screen was TLR7-induced cytokine production, we next asked whether it was IRF5-mediated. This was assessed, in part, through the direct phosphorylation of IRF5. The Reviewer is correct that NetPhos and other prediction tools were not used, as justification of the study was indeed based on known biology downstream of TLRs. Further, few kinase substrates are known for STK25 and thus STK25 is not included in prediction tools such as NetPhos.

The discussion is largely repetitive in that it restates each figure finding prior to discussing it. This should be modified so that the authors discuss some of the more salient aspects of STK25 biology - ie its ability to phosphorylate LATs and what that means in the context of TLR7/8 signaling.

We thank the Reviewer for their suggestion and have shortened the repetitive components of the discussion and include additional perspectives on STK25 and TLR7/8- IRF5 signaling. Regarding LATS and TLR signaling, however, we think there may be some confusion as STK25 phosphorylates large tumor suppressor (LATS) kinases, which are different than the adaptor LAT (linker for activation of T cells) that has been implicated in TLR signaling and undergoes phosphorylation by ZAP-70. To our knowledge, the T cell LAT has not been shown to be phosphorylated by STK25. Thus, this is not included in the Discussion.

Otherwise this is an excellent manuscript that should be accepted.

Thank you!

Sincerely,

Betsy J. Barnes

June 27, 2025

RE: Life Science Alliance Manuscript #LSA-2025-03343-TR

Betsy J Barnes
Feinstein Institute for Medical Research
Center for Autoimmune and Musculoskeletal Diseases
350 Community Dr.
Manhasset 11030

Dear Dr. Barnes,

Thank you for submitting your Research Article entitled "TLR-induced STK25 activation promotes IRF5-mediated inflammation". It is a pleasure to let you know that your manuscript is now accepted for publication in Life Science Alliance. Congratulations on this interesting work.

DISTRIBUTION OF MATERIALS:

Again, congratulations on a very nice paper. I hope you found the review process to be constructive and are pleased with how the manuscript was handled editorially. We look forward to future exciting submissions from your lab.

Sincerely,

Sarita Hebbar, PhD
Scientific Editor
Life Science Alliance
<http://www.lsajournal.org>